# A Novel Hydroforming Process by Combining Internal and External Pressures for High-Strength Steel Wheel Rims

**DOI:** 10.3390/ma15196820

**Published:** 2022-09-30

**Authors:** Wei-Jin Chen, Yong Xu, Hong-Wu Song, Shi-Hong Zhang, Shuai-Feng Chen, Liang-Liang Xia, Yong Wang, Boris-B. Khina, Artur-I. Pokrovsky

**Affiliations:** 1Shi-Changxu Innovation Center for Advanced Materials, Institute of Metal Research, Chinese Academy of Sciences, Shenyang 110016, China; 2School of Materials Science and Engineering, University of Science and Technology of China, Shenyang 110016, China; 3Research Institute, Baoshan Iron and Steel Co., Ltd., Shanghai 201999, China; 4Physical-Technical Institute, National Academy of Science of Belarus, 220141 Minsk, Belarus

**Keywords:** high-strength steel, wheel rim, hydroforming by combining internal and external pressures (HIEP), finite element method (FEM)

## Abstract

As one of the key safety components in motor vehicles, the steel wheel rim is commonly fabricated with the roll forming process. However, due to the varied cross-sections of the rim and the low formability of high-strength steel, it is difficult to produce thin-wall and defect-free wheel rims to realize the purpose of light weight. To solve these problems, a novel hydroforming process by combining internal and external pressures (HIEP) was proposed to produce thin-wall wheel rims in the current study. The designed initial tube with diameter between the maximum and minimum diameter of the wheel rim ensures dispersed deformation and effectively avoids local excessive thinning. During HIEP, a hydroforming process was performed with two successive stages: the external pressure and internal pressure stages. Theoretical analysis and finite element method (FEM) were jointly used to investigate the effect of process parameters on the wrinkling and thinning. With the optimized parameters for internal and external pressure, the wrinkling of wheel rims is prevented under compressive state during the external pressure forming stage. Additionally, HIEP was experimentally carried out with high-strength steel rims of 650 MPa ultimate tensile strength (UTS). Finally, wheel rims with weight reduction of 13% were produced successfully, which shows a uniform thickness distribution with a local maximum thinning ratio of 11.4%.

## 1. Introduction

For lower energy consumption and lower emission of carbon dioxide, a material with a higher strength and a smaller density is expected to be effective in achieving a lightweight vehicle [1,2]. Automobile wheels are basic structural parts of a motor vehicle that support the weight of the vehicle and bear the dynamic stress when driving. Therefore, the weight of the wheels should obviously have a great influence on fuel consumption [3]. There is a significant demand to reduce the weight of automobile wheels.

Aluminum alloy has been widely applied to produce lightweight wheels in the industry for low density [4,5] and nice-looking appearances. Even magnesium alloy wheels have been designed in some high-end motor vehicles for the reduction in weight [6,7,8]. However, aluminum and magnesium wheels produced by the casting process have a high cost and relatively low material strength. Moreover, casting defects, as crucial factors, affect the fatigue property and shock resistance of casting wheels [9]. Thus, aluminum rims are made with large wall thickness to ensure the service performance, which is difficult to achieve lightweight results actually. Compared to as-cast Mg alloys, wrought alloys such as AZ80 magnesium have shown better ductility and fatigue properties [10,11], whereas, compared with aluminum alloy and magnesium alloy, higher strength steel is more cost-effective to reduce the weight of the wheels [12]. The steel wheel is mostly composed of disc and rim welded together. Hot-rolled dual-phase steel with 700 MPa UTS has been tried for manufacturing the disc by a multi-stage stamping process to reduce weight [13]. The rim of steel wheel is usually cold-formed through a multi-stage manufacturing process, as shown in Figure 1. Due to the high effectiveness, the rolling process is a common operation to manufacture rims with cross-sectional shapes. However, cold forming requires the well formability of steel. Actually, many defects occur during current manufacturing process, such as local excessive thinning or cracking and poor surface quality with scratches and abrasions [14,15,16]. Steels with 800 MPa ultimate tensile strength have been tried for the manufacture of wheels, yet steel with 600 MPa UTS is the preferred material for rim application [17,18]. Meanwhile, the rim directly bears the most force from the automobile body, and the forming quality of the rim directly affects the service performance of the wheel [19,20]. The current rim manufacturing process has high requirements on the formability of material, which is the main problem limiting the lightweight application of higher strength steel.

The tube hydroforming (THF) technology has been widely used to manufacture hollow components in the automobile and aerospace industries [6,21,22]. Due to hydraulic forming process, the surface quality of the component can be much better than that using a rigid punch. It seems to be possible to produce steel wheel rims through THF. However, in a conventional tube hydroforming process, the initial tube diameter must be the minimum diameter of the component, and it is common to use a high hydraulic pressure inside the tube blank to expand the metal into the required shape with the die cavity. High pressure amplifies the influence of friction and makes it difficult for the material to fill the corner, which may result in rupturing and excessive thinning, especially for components with varied cross-sections or a small corner radius [23,24]. The wheel rim typically has a hollow characteristic with varied cross-sections, so it is difficult to produce a perfect wheel rim without cracking or excessive thinning by the conventional tube hydroforming process.

Many methods of the THF process have been proposed to improve cracking or excessive thinning and to decrease the hydraulic pressure for hollow components. Movable bushes have been commonly used in tube hydroforming to form corners of small radius [25,26,27]. The corner is deformed under internal pressure and movable bushes. The tensile stress state of corner is transformed to compressive stress state, which results in a better corner filling under lower hydraulic pressure than conventional dies. However, it mainly focuses on the improvement of local thinning and cracking. Tube hydro-forging is based on hydroforming and makes use of the force from movable dies, which is similar to tube hydro-mechanical forming [28,29]. In this process, the small corner is deformed under compressive stress situation of the rigid die, while the role of the internal pressure is translated into the function of support. However, the tube was formed under rigid die movement, which may affect the surface quality. Double-sided hydroforming has been used in production because it can significantly improve the formability and surface quality of sheet metals [30,31,32]. The internal and external hydraulic pressure are employed simultaneously and the internal pressure or the external pressure is used as the back pressure to increase the hydrostatic stress value, which enhances the formability of the components. However, it is still a challenge to form thin-walled components with complicated structures and large deformation for material of low formability.

In the present study, hydroforming through combination of internal and external pressures (HIEP) was proposed to produce high-strength steel wheel rims. The principle of this method is introduced. Theoretical analysis and finite element method (FEM) were jointly used to investigate the effect of process parameters on the wrinkling and thinning. With the optimized parameters during HIEP, the deformation can be greatly dispersed to reduce the requirement of material formability so as to adopt higher strength steel for lighter weight. Finally, the proposed method is applied for the actual manufacturing of steel rims. The result showed that the component has high forming quality, and the feasibility of the method was proven.

## 2. Process Principle

### 2.1. Principles of HIEP

The principle of HIEP analyzed in this study is shown in Figure 2. In HIEP, the primary parameter is the diameter of initial tube which can be determined between the maximum and minimum diameter, as shown in Figure 2a, which is distinctly different from conventional hydroforming process. Then, the internal pressure, P_i_, and external pressure, P_e_, respectively, played the role of forming metal to fill the die cavity. Part of the tube is formed to the target shape of mandrel under the external pressure in the compressive zone, while the other part of the tube is bulged under the internal pressure, as depicted in Figure 2b.

It can be seen that part of the component is formed under a compressive state in the external pressure forming stage, which not only decreases the risk of excessive thinning at that region but also decreases the expansion amount at the other region. The deformation can greatly disperse in this new process. As result, the risk of excessive thinning and cracking of the total component can be reduced inevitably. Moreover, as the component is formed by the liquid during the total process, the surface quality of components can be maintained well.

Obviously, in this novel process, the initial tube diameter is the most critical parameter. However, there is a compressive stress state along the radial direction of the tube blank under the external pressure. In this case, the risk of wrinkling should be a concern.

### 2.2. Wrinkling Criterion of the Thin-Wall Component under an External Pressure

For an element at the middle of this tube surface, the stress state under the external pressure is shown in Figure 3. Because the thickness is far less than the tube diameter, the plane stress condition is assumed, where $\sigma_{\theta }$ and $\sigma_{z}$ are the stress components in the circumferential and axial directions, $\rho_{\theta }$ and $\rho_{z}$ are the circumferential and longitudinal radii, *t* is the tube thickness, and D and R are the tube diameter and radius. Then, the force equilibrium equation can be written as:(1)$$\frac{\sigma_{z}}{\rho_{z}}+\frac{\sigma_{\theta }}{\rho_{\theta }}=\frac{P_{e}}{t}$$

Using the Hill yield criterion, the equivalent stress can be described as:(2)$$\bar{\sigma }=\sqrt{\sigma_{\theta }^{2}+\sigma_{z}^{2}-\frac{2r}{1+r}\sigma_{\theta }\sigma_{z}}$$
where *r* is the normal anisotropy coefficient.

According to the Levy–Mises flow rule (assuming a constant volume), the plastic strain increment along the thickness direction can be written as:(3)$$d\varepsilon_{t}=-\frac{d\bar{\varepsilon }}{2\bar{\sigma }}\left(\sigma_{\theta }+\sigma_{z}\right)$$
where $d\bar{\varepsilon }$ is the equivalent plastic strain increment.

When the end of the tube is free, $\rho_{\theta }=0.5D=R$, $\rho_{z}=\infty$. Combining Equations (1) and (2), the critical yield pressure under the external pressure condition $P_{s}$ can be written as:(4)$$P_{s}=\frac{2t\sigma_{s}}{D\sqrt{1-\frac{2r}{r+1}\alpha +\alpha^{2}}}$$
where α is the ratio of the principal stresses (ratio of the circumferential stress to longitudinal stress), which can be described as:(5)$$\alpha =\frac{\sigma_{z}}{\sigma_{\theta }}$$

$\sigma_{s}$ is the critical yield stress of the material. The critical elastic buckling pressure $p_{cr}$ and critical plastic buckling pressure $p_{cr-p}$ can be calculated by [33]:(6)$$P_{cr}=\frac{0.883a_{p}E{\left(2t/D\right)}^{2.5}}{L/R}$$
(7)$$P_{cr-p}=P_{s}+\frac{0.883a_{p}E_{t}{\left(2t/D\right)}^{2.5}}{L/R}$$
where *E* is the elastic modulus, $E_{t}$ is the tangent modulus, *L* is the length of the compressive area, $a_{p}$ is a constant associated with parameter Z of S B Bathorf, which is approximately equal to 1.51.
(8)$$P_{cr}>P_{s}$$

Equation (8) is the base condition to determine the tube size and compressive process. Then the initial tube diameter and short compressive region length can be calculated by combining Equations (4), (6), and (8) as:(9)$$DL^{2}<\frac{1.56t^{3}a_{p}{}^{2}E^{2}\left(1-\frac{2r}{r+1}\alpha +\alpha^{2}\right)}{\sigma_{s}{}^{2}}$$

Equation (9) can be used to determine the initial tube diameter from critical elastic buckling pressure. Obviously, a small initial tube diameter and short compressive region length should be primarily considered to decrease the wrinkling risk.

Generally, wrinkling happens easily in the process of compressive deformation, mainly due to the local accumulation of materials and thickening. Therefore, the trend of material thickening can be judged by the variation of surface area based on the principle of constant volume.

As shown in Figure 4, the compressive deformation shape is simplified as a dumbbell shape, and the transition area is set as a straight line. Assuming that no material flows into the deformation region from the outside, the surface area of the initial tube corresponding to the deformation area is *S*_0_, and the area of the deformation region can be divided into *S*_1_ at the bottom of the dumbbell and *S*_2_ at the transition region. The angle between the slope and the bottom of the deformation region is defined as *θ*. *x* is the vertical distance between the bottom plane edge of the deformation region and the slope region. Then, the maximum *x* is calculated as follows:(10)$$x_{max}=\frac{D_{0}-D_{1}}{tan\theta }$$

The surface area of the initial tube corresponding to the deformation area can be calculated as:(11)$$s_{0}=\pi *D_{0}*L$$

The bottom area of the dumbbell can be obtained as follows:(12)$$s_{1}=\pi *D_{1}*\left(L-2x_{max}\right)$$

A differential dy on the slope is taken out, which corresponds to *dx* at the bottom. The height is defined as *h*, and then dy and *h* can be written as:(13)$$dy=\frac{dx}{cos\theta }$$
(14)$$h=x*\tan\theta +D_{1}$$

Then, the slope area is described as:(15)$$s_{2}=2\int_{0}^{x_{max}}\pi *h*dy$$

$S_{0}^{\prime }$ is defined as the area of the component reducing region. Combining Equations (10)–(15), $S_{0}^{\prime }$ can be calculated by:(16)$$S_{0}^{\prime }=S_{1}+S_{2}=\pi *D_{1}*\left(L-2*\frac{D_{0}-D_{1}}{\tan\theta }\right)+2*\frac{\pi }{\sin\theta }*\left(D_{0}^{2}-D_{1}^{2}\right)$$
(17)$$S_{0}^{\prime }\geq S_{0}$$

Equation (17) can be used to determine the initial tube diameter from the thickening trend.

## 3. Preparation

### 3.1. Component

The geometry of the wheel rim in this study is presented in Figure 5. It has typical varied cross-sections, with a maximum diameter of 389 mm and a minimum diameter of 318 mm. There are several stiff walls in the zones of the rim slot and rim flange, which makes it difficult to control the thinning under the roll-forming process. The rim slot is welded directly with the disc to bear the weight of the vehicle frame. The rim flange is used to constrain the tire. As a safety component, the bearing capacity and fatigue property of the wheel rim are most important. Obviously, the excessive thinning at the critical zone must affect the rim service performance. When higher steel with a lower formability is used to decrease the sheet thickness of the rim, it is even more difficult to form through the conventional cold roll-forming process. In brief, it is vital to achieve a uniform thickness distribution for the lightweight rim.

### 3.2. Material

In the current study, hot-rolled high-strength low-alloy steel of 650 MPa UTS was employed to produce the wheel rim through new process of HIEP. The sheet thickness of 2 mm was adopt to upgrade from the 540 MPa UTS wheel rim of 2.3 mm thickness so as to achieve at least 13% weight reduction. The sheets, which were supplied by Baosteel, were rounded, and welded to become tubes. The material chemical compositions and mechanical properties are listed in Table 1 and Table 2.

### 3.3. Finite Element Model

To investigate the process of hydroforming for the wheel rim, the finite element model was developed with Abaqus/Explicit. The dies were modeled as an analytical rigid body. Meanwhile, the tube blank was discretized by elastic–plastic quadrilateral shell elements with the element type of S4R. The Young’s modulus is 207 GPa, while the Poisson’s ratio is 0.28. The stress–strain model used in this study was fitted from the test results of the material B650CL. An isotropic plasticity model was applied in the simulation. The friction coefficient between the tube and die was set at 0.1. In order to balance computational efficiency and accuracy, the mass scaling factor in this model is set to 100 to ensure the ratio of kinetic energy to internal energy is less than 5%.

Considering the simplicity of the die design and equipment usage, the schematic of the forming process was divided into two forming stages in the current study: the external pressure and internal pressure hydroforming stages. As shown in Figure 6, firstly, the external pressure forming stage was used to form the tube to achieve the shape of the mandrel in the rim slot zone under the external pressure. Then, another set of dies was used to form the tube to expand the final shape of the rim under the internal pressure. Meanwhile, the movable dies of bush 1 and bush 2 were employed to improve the corner filling.

### 3.4. Experimental Setup

The experimental setup and dies of the HIEP are shown in Figure 7. The 315 t press and 1500 t press were adopted to form the rim during the external pressure forming stage and internal pressure forming stage, respectively. The dies in the external pressure forming stage consisted of mandrel and seal parts, as shown in Figure 7a. In order to simplify the experimental equipment, the internal pressure forming process was decomposed into two sequences, as shown in Figure 7b,c. Thus the simultaneous axial feeding of both ends was replaced by feeding one after another. The dies in the internal pressure forming stage consisted of the die, fixed part, and bushes. The shapes of the movable bushes were designed as part of the die, which were employed to further improve the uniformity of the thickness distribution.

## 4. Results and Discussion

### 4.1. Conventional Hydroforming Process for Rim

Firstly, the conventional hydroforming process with an internal hydraulic pressure was employed to manufacture the wheel rim using the FEM. The diameter of the tube blank here is determined with the minimum diameter of 318 mm of the wheel rim. The process is designed as shown in Figure 8.

The moveable bushes are employed to improve the corner filling. During the pre-hydroforming step, the tube is pre-bulged to a transitional shape under a low internal hydraulic pressure. Then, the movable bushes move forward to the final position to force the material to fill the corner of the die under the support of the internal pressure. The best loading path can be determined with the finite element simulation. However, it is difficult to produce a perfect component without cracking or excessive thinning. As shown in Figure 9, cracking would still occur in the flange region and the internal hydraulic pressure of 200 MPa needs to be employed to fill the corner in the calibration step. As shown in Figure 10, the minimum thickness is 1.377 mm. It means that the local thinning ratio of the component is over 31%, which is beyond the limit of cracking and not permitted by industrial application.

### 4.2. Parameters on Wrinkling during the External Pressure Stage of HIEP

#### 4.2.1. The Influence of Initial Tube Diameter

In the process of HIEP, the initial tube diameter can be larger than the minimum diameter of the wheel rim (318 mm). In order to minimize the amount of bulging deformation in the internal pressure forming stage, it is better to choose the initial tube with a larger diameter. However, the amount of deformation in the compressive forming process is mainly determined by the initial tube diameter. Smaller diameter of the initial tube is better for controlling the wrinkling during the diameter reducing process.

The critical external elastic buckling pressure $p_{cr}$ and the critical tube yield pressure $p_{s}$ can be calculated through Equations (4) and (6). Then, the initial tube diameter and compressive region length can be calculated by Equation (9). When the initial tube diameter and the diameter reduction region length become larger, $p_{cr}$ will be smaller, which means there is a greater risk of instability. As shown in Figure 11, when the initial tube diameter reaches 390 mm, the $p_{cr}$ is even less than the critical tube yield pressure $p_{s}$ and the elastic instability will occur in the elastic stage. Therefore, the initial tube diameter should be smaller than 390 mm.

Meanwhile, the trend of material thickening is calculated by the variation of surface area to determine the initial tube diameter. From the 3D model of the wheel rim, the area of the component reducing region is measured to be 80,026.6 mm^2^. Assuming the two sides of the tube are restrained, the initial surface area corresponding to the reducing region with the diameter of 340 mm is calculated to be 79,928.9 mm^2^ by Equation (11), which is just smaller than the area of the reducing diameter region. According to Equation (17), the diameter should be smaller than 340 mm.

In summary, to balance the decreasing bulging deformation and wrinkling tendency, the initial tube diameter of 340 mm is determined for the rim hydroforming in this study. More precisely, the compressive deformation under the action of the external pressure inevitably induces the material to flow non-uniformly; therefore, local wrinkling may still occur in the compressive deformation region.

#### 4.2.2. The Influence of the Value of External Pressure

During the first stage of HIEP, as shown in Figure 6, the central region of the initial tube is compressively formed to reduce the diameter under the external pressure along the radial direction. In contrast with the tube deformation under the internal pressure, the forming region of the tube actually bears compressive stress under the external pressure, which easily induces a buckling instability. The compressive deformation along the radial direction and equivalent plastic strain of the tube under different external pressure values are shown in Figure 12. When the pressure exceeds 7 MPa, the tube enters the state of plastic deformation from the state of elastic deformation, as shown in Figure 12a,b. From Figure 12c, it is obvious that the tube can reduce the diameter uniformly along the radial direction in the early stage of plastic compression, and there is no wrinkling around the tube. Figure 12d–f shows that the central region of the tube enters the state of unstable deformation as the external pressure increases to 12 MPa.

Tube instability occurs in the plastic deformation region during the external pressure hydroforming stage. In the early stage of the circumferential instability, there is significant petal buckling. ∆H between the high point and low point of the adjacent wave is used to characterize the variation of the instability height under the external pressure. The maximum height difference ∆H between the wave peak and wave valley may reach 2–5 mm (Figure 13). As the deformation continues, the total region of the tube along the circumferential direction goes in the compressive stress state. The external pressure plays the role of calibration during final stage of plastic deformation. When the wave peak gradually is forced to be closed to the mandrel, the maximum height difference ∆H decreases. Meanwhile, there is a significant tendency of thickening at the wave peak region. As shown in Figure 14, the tube thickness at the wave peak region is larger than that at wave valley. The larger the value of external pressure in the range of 0–60 MPa, the lower tendency of instability height can be obtained.

#### 4.2.3. The Influence of Material Flowing Displacements

Actually, the stress state at the reducing diameter region is not invariable during the different compressive deformation periods. The tube is in contact with the mandrel only by friction. Figure 15 shows the axial stress variation under a typical external pressure. Point A represents the bottom of the reducing diameter region, point B represents the corner region, and point C represents the edge of the reducing diameter region. At the initial stage of plastic deformation, the tube in the reducing diameter region all bears the tensile stress, while the other region outside bears the compressive stress. It illustrates that the material outside the deformation region is flowing into the compressive deformation region at this time. When the reducing diameter region is forced to contact the mandrel, the tube mostly bears compressive stress due to the surplus material flowing. Therefore, the instability and thickening tendency of the materials are mainly closely related to the material flowing into the deformation zone.

According to Equation (3), the thickening trend of the tube thickness can be estimated by the sum of $\sigma_{\theta }$ and $\sigma_{z}$. Because the circumferential compressive stress is greater than the axial tensile stress under compression, so there must be thickening in radial direction. However, it is helpful to increase the axial tensile stress, which can reduce the thickening trend. Figure 16 shows the variation of the instability under different material flowing displacements from 0 to 3 mm. Obviously, the ∆H of the high point and low point of the adjacent wave decreases as the material flowing displacement is limited or fixed, which illustrates that the instability tendency should be reduced by controlling the material flowing. Simultaneously, the thickening distribution in the compressive region is also developing when the material is restricted to flow into the deformation region, which is obvious as shown in Figure 17. The maximum thicknesses of points A, B, and C are decreasing due to the decrease of material flowing displacement, as shown in Figure 18. Therefore, it is necessary to limit, even fix, the material flowing displacements for controlling wrinkling under external pressure.

### 4.3. Critical Loading Path during the Internal Pressure Stage of HIEP

The second stage of HIEP actually can be regarded as a conventional hydroforming process. The compression-formed tube after the first stage was employed to bulge during this stage to obtain the final component shape. For the complex cross-section shape of the rim flange, it is still difficult to fill the flange region with a low thickness thinning ratio. The moveable bushes are also employed to improve the corner filling and to reduce the internal hydraulic pressure.

As shown in Figure 19, the internal pressure stage actually consists of three steps: pre-hydroforming, movable bush punching, and calibration. At the beginning of the internal pressure forming stage, the movable bush is placed a certain distance from the final position. During the pre-hydroforming step, the tube is pre-bulged to a transitional shape under a low internal hydraulic pressure, while the movable bushes are kept stationary. Then, the movable bushes move forward to the final position to force the material to fill the corner of the die under the support of the lower internal pressure. At this moment, the pressure only plays the role of a support to avoid tube wrinkling or folding during the step of the movable bushes moving forward. Thereafter, the calibration step is carried out in which the internal hydraulic pressure continues to increase to force the tube to fully fill the die cavity. Obviously, the material can easily flow into the flange region, which improves the thickness distribution due to pre-hydroforming and the moving of the movable bush.

The matched combination of the internal hydraulic pressure with the movable bush moving displacement is the key parameter that can be optimized and determined through the finite element simulation. As shown in Figure 20, the loading path 2 is designed to obtain no cracking and a low plastic strain distribution during the rim expansion process with 100 MPa pressure. Under the constant internal pressure of 15 MPa during the step of the movable bushes moving forward, as with path 1, when the bush 1 moving displacement is less than 22 mm or the bush 2 moving displacement is less than 18 mm, cracking will occur in the transition zone and there will be a lack of material in the rim flange region. In addition, as with the loading path 3, when the bush 1 moving displacement exceeds 22 mm or the bush 2 moving displacement exceeds 18 mm, there will be material folding in the rim flange region. The thickness distribution under loading path 2 is also more uniform than the others, as shown in Figure 21. The minimum thickness of rim under loading path 2 is 1.766 mm. It means that the largest thinning ratio is less than 12%, which is much lower than that in the conventional hydroforming process.

### 4.4. Experimental Verification

With the guidance of theoretical analysis and FEM, HIEP was experimentally carried out. Firstly, the initial tube diameter of 340 mm was adopt and the tubes were formed under external pressure. The external pressure was loaded linearly from 0 to 100 MPa. Without constraint design in the original die1, there is only friction relationship between the surfaces of tube and die1. Wrinkling indeed occurred at the compressive region while the material flowing displacement was about 3 mm. As shown in Figure 22, the morphology of wrinkling is in good agreement with the simulation result. According to the concept of controlling the material flowing displacement, the die was optimized. As shown in Figure 23a, the seal part was divided into two parts: the seal part and the binder holding part. The binder holding part consists of two semi-rings which can provide holder force by bolts. Furthermore, welding spots were designed and placed at the inner surface of the seal part, which can increase the clamping force. Due to these optimizations of the die, the material flowing displacement was limited to less than 0.5 mm in the experiments. As a result, the wrinkling was effectively improved, as shown in Figure 23b.

Then, the tubes formed by external pressure hydroforming were further adopted to deform under internal pressure hydroforming. In the pre-hydroforming step, the internal pressure increased linearly from 0 MPa to 20 MPa as the bushes remained motionless. Then, the moveable bushes moved forward to provide axial feeding while the internal pressure held a constant value of 15 MPa to play the role as support. When the bushes reached close position, the internal pressure increased to 100 MPa to calibrate. As shown in Figure 24a,b, there will be a lack of material or folding in the rim flange region when the movable bush stroke does not match with the internal pressure. With the bush 1 moving displacement 22 mm and the bush 2 moving displacement 18 mm under the support pressure value of 15 MPa, the perfect component with the desired shape was produced to match well with the die, as shown in Figure 24c–e, and the local characteristics were completely clear under the pressure value of 100 MPa. After the surface was painted, the lightweight rim and wheel were produced completely and successfully, as shown in Figure 24f. From Figure 25, it can be seen that the thickness distribution in the FEM results of the rim is uniform with a maximum thickness of 2.08 mm and a minimum thickness of 1.78 mm. Then, the experimental rim was cut to measure the thickness distribution (maximum thickness of 2.04 mm and minimum thickness of 1.73 mm). Thus, the maximum thinning ratio on the experimental rim is 11.4%, which is close to the simulation result of 11.0%. It can be observed that the most extreme relative error of the wall thickness is just 3.6%, contrasting the trial and numerical outcomes.

## 5. Conclusions

In this work, to solve the forming problems during manufacturing lightweight steel wheel rims, the process of hydroforming by combining internal and external pressures (HIEP) was proposed. The thin-wall wheel rim of 650 MPa (UTS) high-strength steel was successfully produced by this new process to achieve at least 13% weight reduction. The main conclusions can be summarized as follows:(1)During HIEP, the hydroforming process was performed with two successive stages: the external pressure and internal pressure stages. The internal and external pressure were employed and respectively played the function of forming in this new process.(2)In contrast with the conventional hydroforming process, the designed initial tube with diameter between the maximum and minimum diameter of the component ensures dispersed deformation, which can effectively enhance the uniform deformation and reduce the minimum pressure required for calibration.(3)Based on theoretical analysis and FEM, the initial tube diameter, deformation region length, and material flowing displacement are the key parameters to prevent wrinkling under the compressive forming state, which were verified by experiments.(4)With the optimized parameters for internal and external pressure stages, the steel wheel rim with varied cross-sections was successfully manufactured without cracking or wrinkling by HIEP under a maximum bulging pressure of 100 MPa. The maximum thinning ratio was only 11.4%, which was much lower than the value for a rim manufactured with the conventional hydroforming process.

## Figures and Tables

**Figure 1 materials-15-06820-f001:**
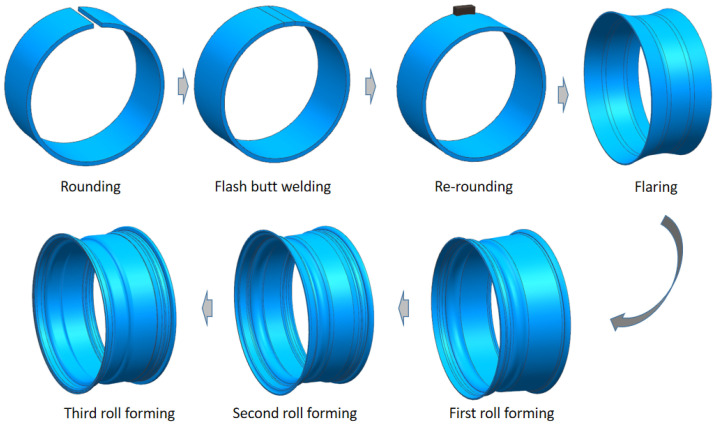
Conventional forming process of steel wheel rim.

**Figure 2 materials-15-06820-f002:**
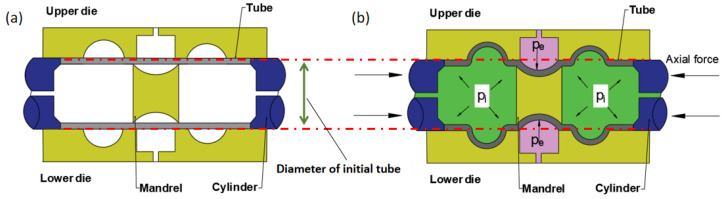
Principle of HIEP: (**a**) before HIEP and (**b**) after HIEP.

**Figure 3 materials-15-06820-f003:**
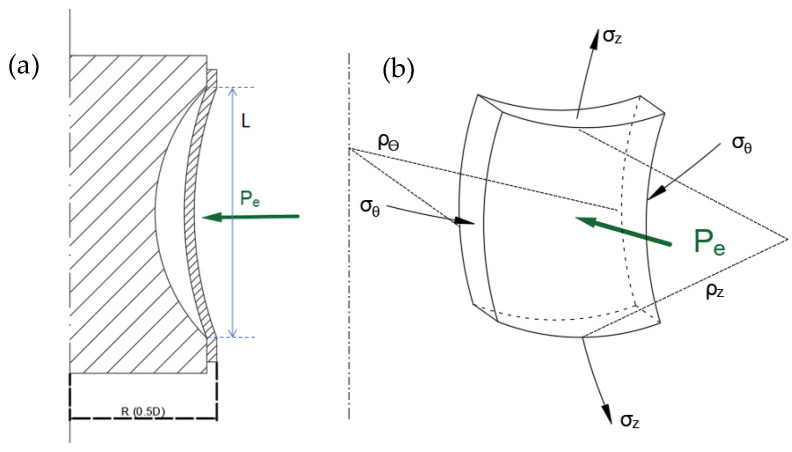
(**a**) Schematic view of compressive deformation under an external pressure and (**b**) stress state of an element of the tube at the compressive region.

**Figure 4 materials-15-06820-f004:**
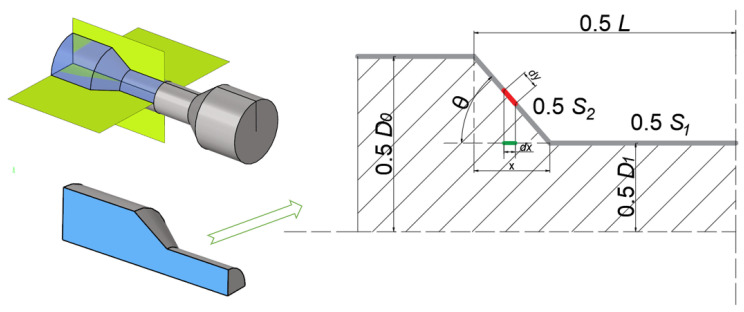
The simplified shape of the compressive region of the tube.

**Figure 5 materials-15-06820-f005:**
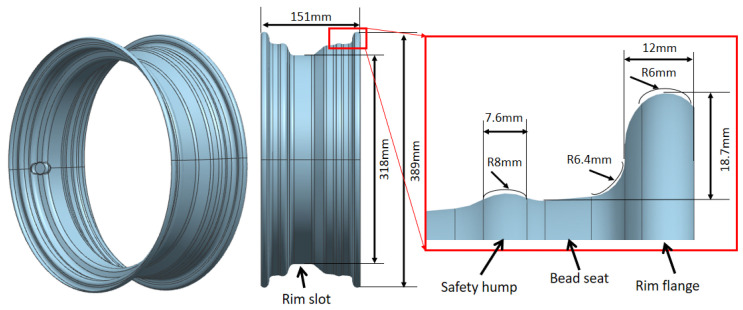
The 3D model of the wheel rim.

**Figure 6 materials-15-06820-f006:**
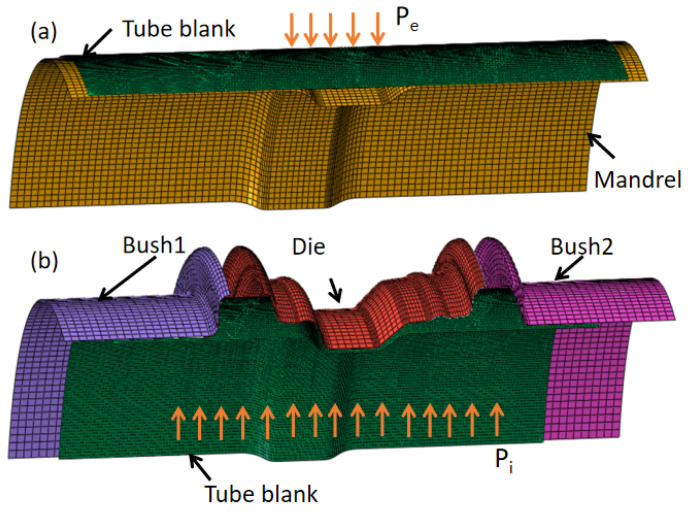
Process schematic of HIEP in the current study: (**a**) external pressure forming stage and (**b**) internal pressure forming stage.

**Figure 7 materials-15-06820-f007:**
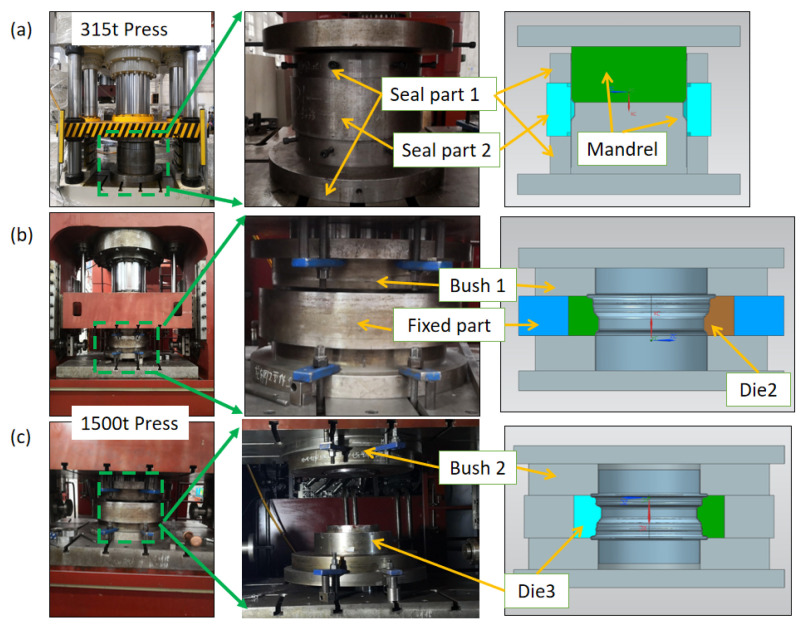
Experimental setup and dies: (**a**) external pressure forming stage, (**b**) one side of internal pressure forming stage, and (**c**) other side of internal pressure forming stage.

**Figure 8 materials-15-06820-f008:**
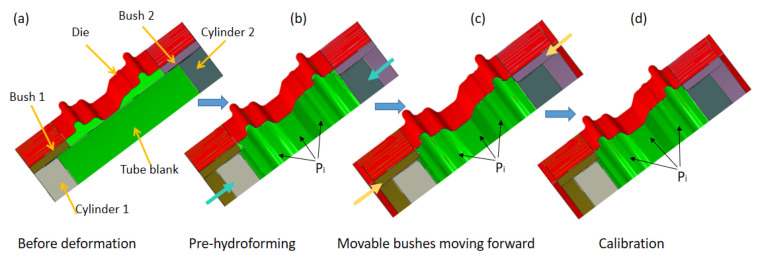
The process design with the conventional hydroforming process: (**a**) before deformation, (**b**) pre-hydroforming, (**c**) movable bushes moving forward, and (**d**) calibration.

**Figure 9 materials-15-06820-f009:**
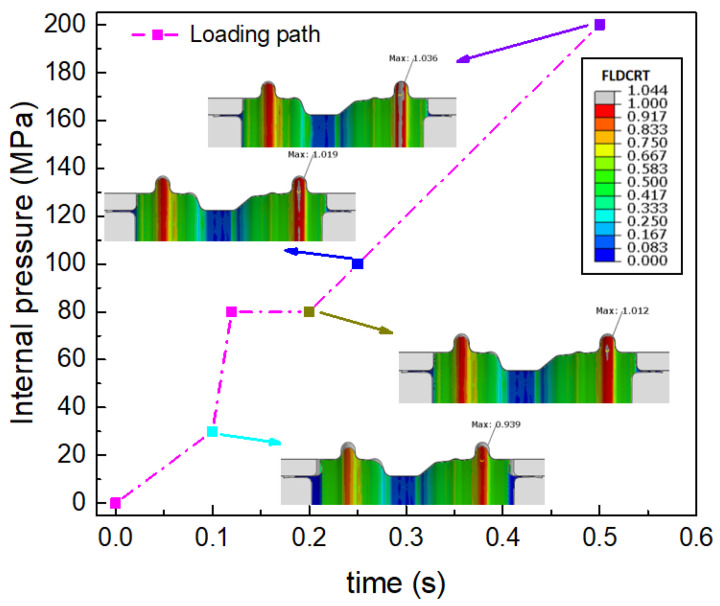
The forming limit results with the conventional hydroforming process. (FLDCRT > 1 means the strain out of material forming limit).

**Figure 10 materials-15-06820-f010:**
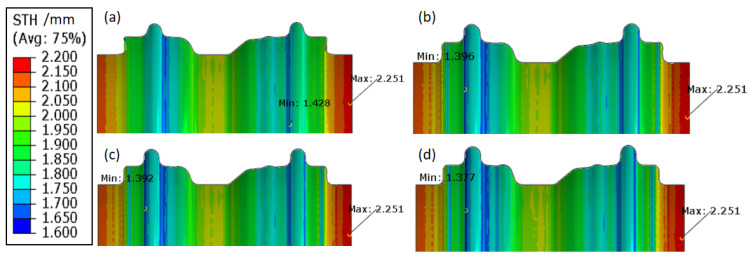
The thickness distribution at different steps: (**a**) after pre-hydroforming, (**b**) finish time of the movable bushes movement, (**c**) 100 MPa internal hydraulic pressure in the calibration step, and (**d**) 200 MPa internal hydraulic pressure in the calibration step.

**Figure 11 materials-15-06820-f011:**
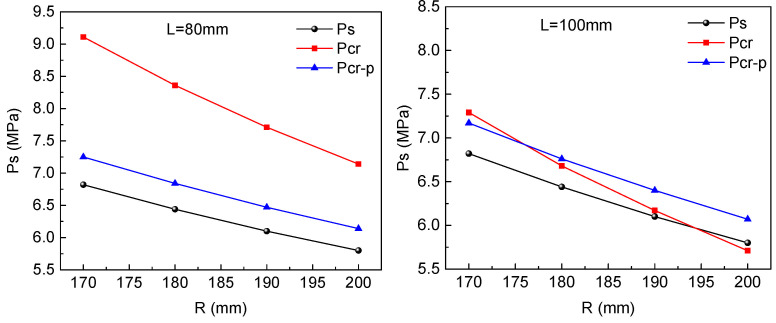
Effect of the initial tube diameter and length of the reducing diameter region on the critical external buckling pressure.

**Figure 12 materials-15-06820-f012:**
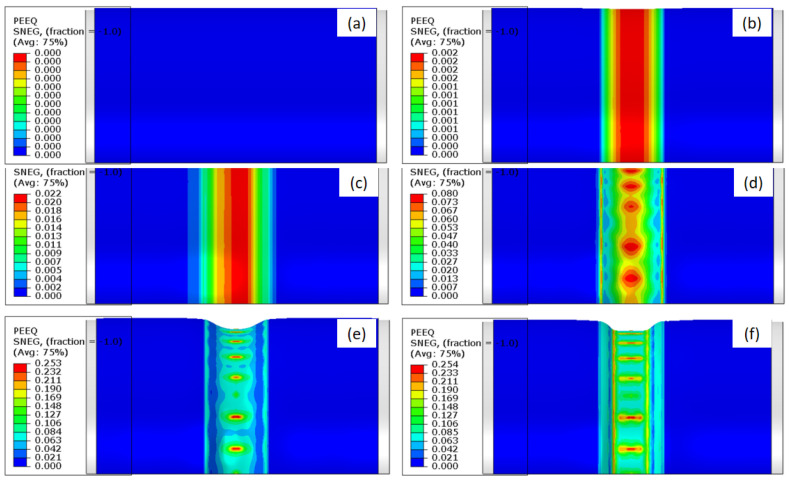
The variation of the equivalent plastic strain distribution under different external pressure values: (**a**) 0 MPa, (**b**) 7 MPa, (**c**) 10 MPa, (**d**) 12 MPa, (**e**) 20 MPa, (**f**) 100 MPa.

**Figure 13 materials-15-06820-f013:**
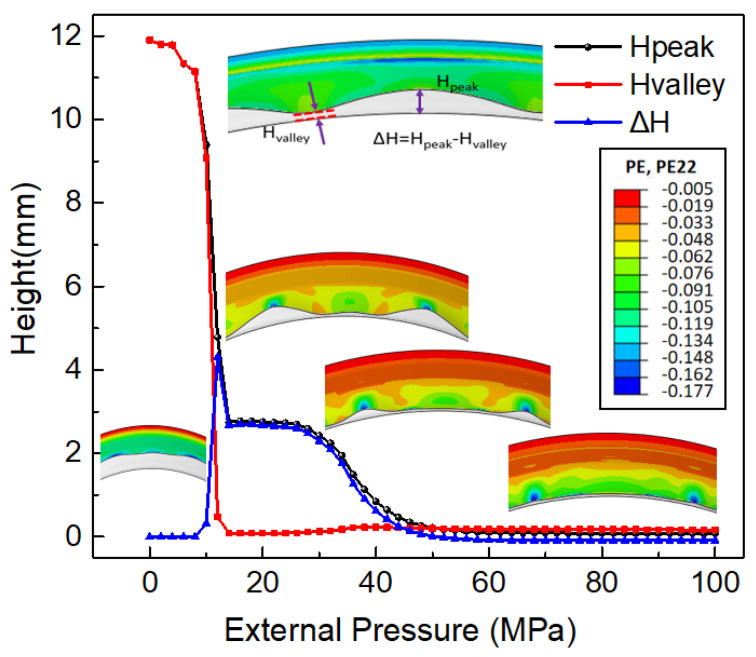
The influence of the external pressure on the instability height.

**Figure 14 materials-15-06820-f014:**
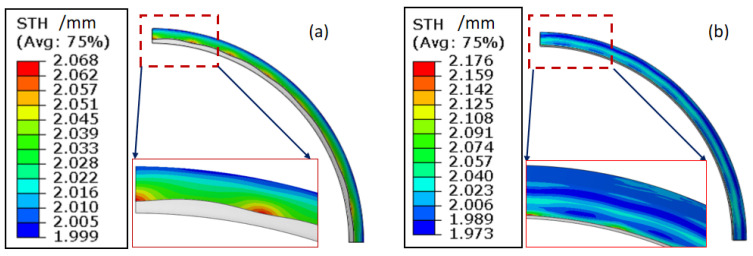
The thickness distribution of the compressive deformation region during different periods: (**a**) initial time of deformation and (**b**) end time of deformation.

**Figure 15 materials-15-06820-f015:**
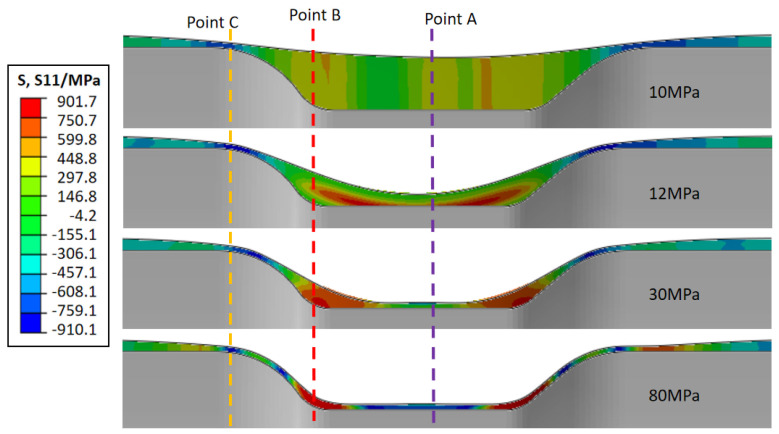
The axial stress variation during compressive deformation.

**Figure 16 materials-15-06820-f016:**
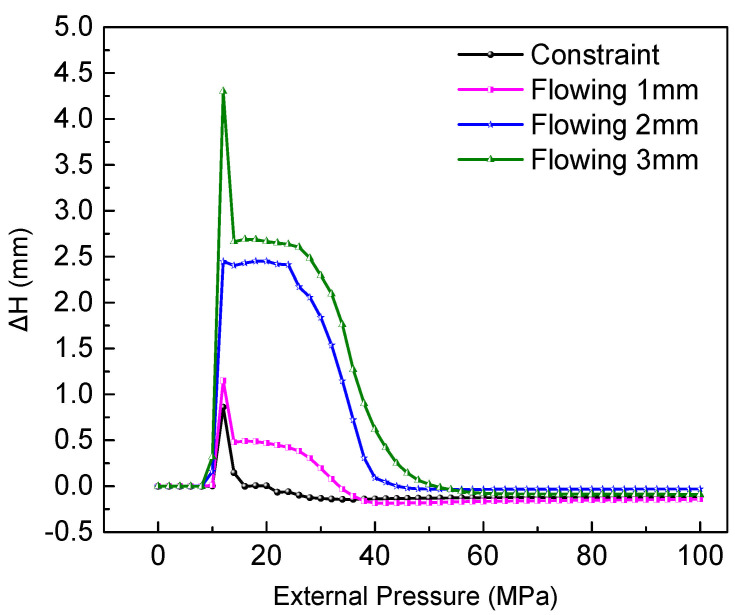
Variation of ∆H under different material flowing displacements.

**Figure 17 materials-15-06820-f017:**
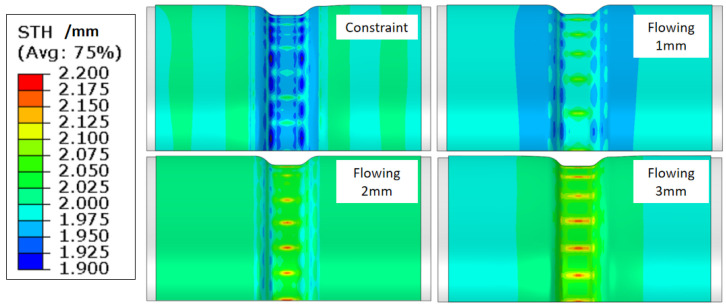
Variation of thickness distribution under different material flowing displacements.

**Figure 18 materials-15-06820-f018:**
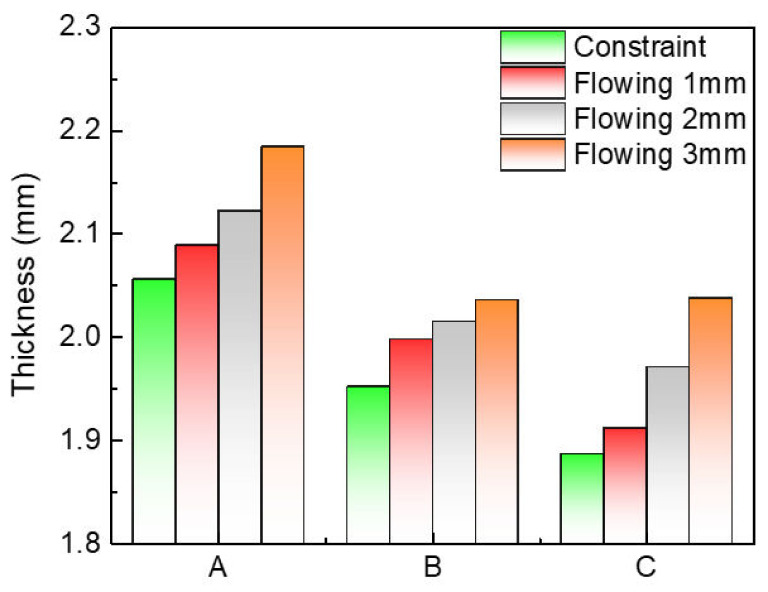
The thickness of point A, B, and C under different material flowing displacements.

**Figure 19 materials-15-06820-f019:**
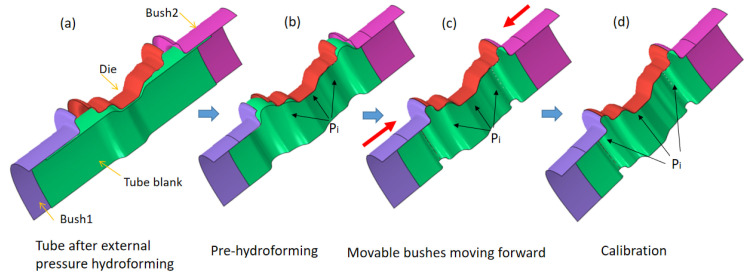
The process of the internal pressure stage with movable bushes: (**a**) before deformation, (**b**) pre-hydroforming, (**c**) movable bushes moving forward, and (**d**) calibration.

**Figure 20 materials-15-06820-f020:**
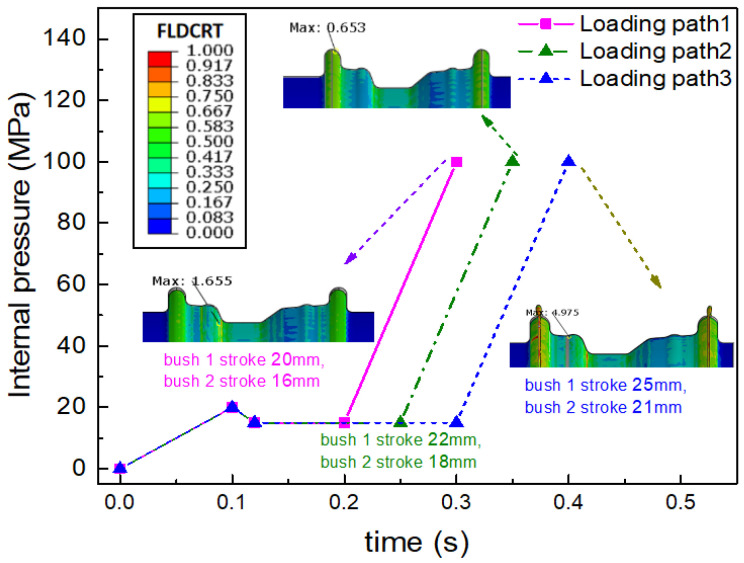
The forming limit results under different loading paths.

**Figure 21 materials-15-06820-f021:**
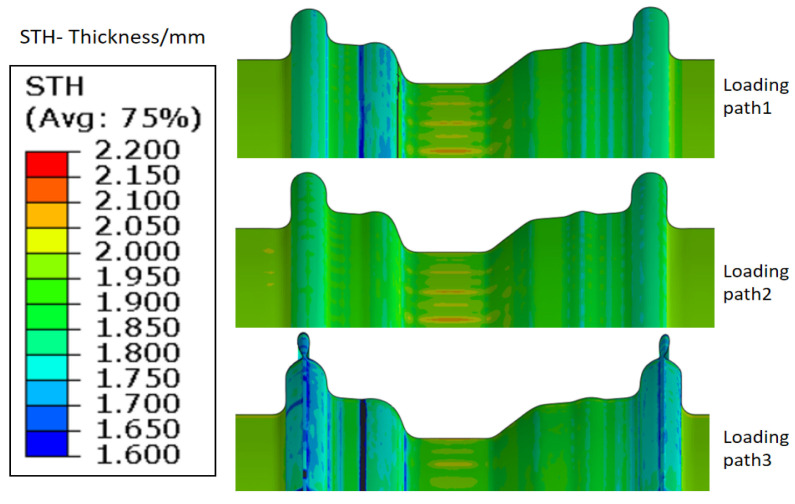
Thickness distribution of components under different loading paths.

**Figure 22 materials-15-06820-f022:**
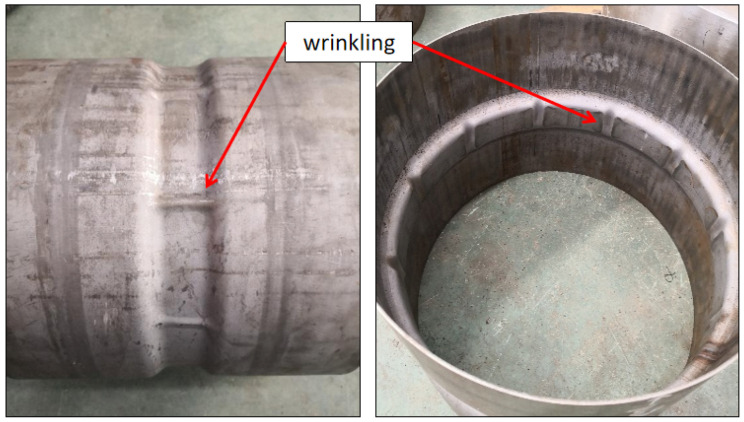
Wrinkling in experiments under the external pressure forming stage.

**Figure 23 materials-15-06820-f023:**
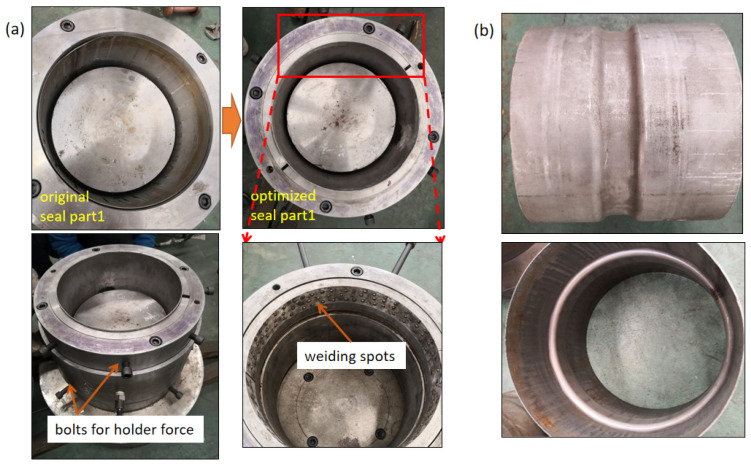
The improvement of controlling material flowing: (**a**) die optimization and (**b**) fine component.

**Figure 24 materials-15-06820-f024:**
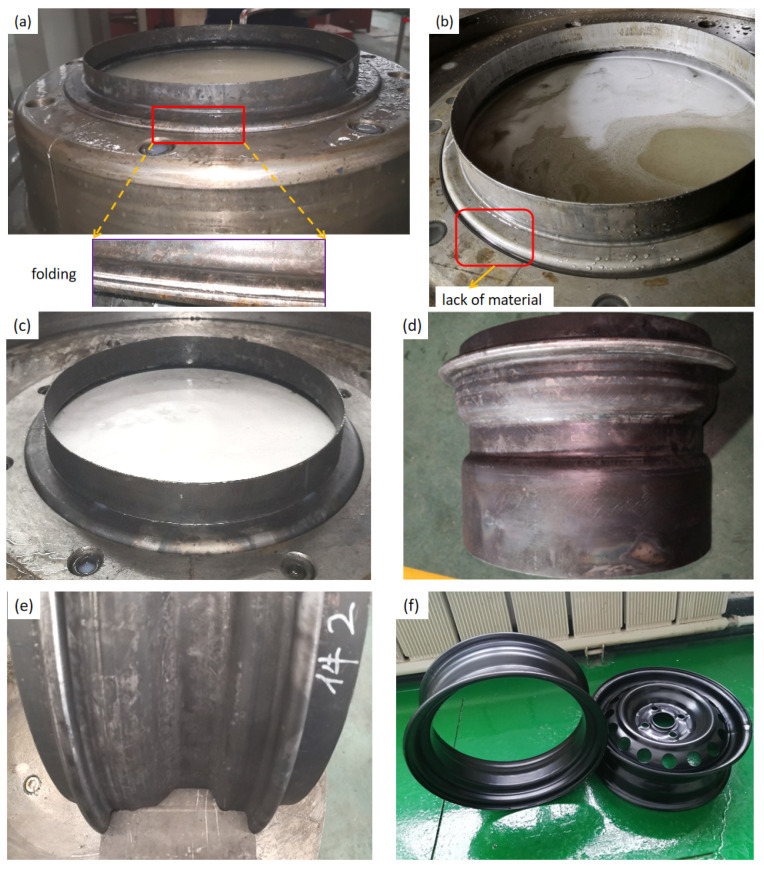
Experimental results: (**a**) material folding in the flange region, (**b**) lack of material in the flange region, (**c**) perfect in the flange region, (**d**) the rim deformed after axial feeding at one end, (**e**) the perfect rim matched well with the die, and (**f**) final rim and wheel.

**Figure 25 materials-15-06820-f025:**
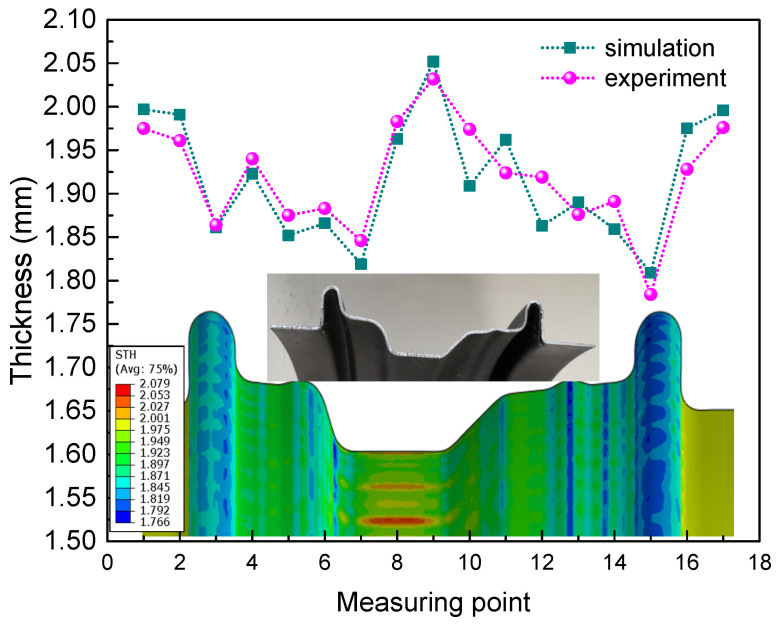
Comparison between the simulation thickness distributions and experimental results along the axial direction of the rim.

**Table 1 materials-15-06820-t001:** Chemical compositions of test material w/%.

Steel	C	Si	Mn	P	S	Ti
B650CL	0.05~0.09	≤0.02	1.0~1.4	≤0.01	≤0.006	0.07~0.09

**Table 2 materials-15-06820-t002:** Mechanical properties of test material.

Steel	Yield Strength (YS)/MPa	Ultimate Tensile Strength (UTS)/MPa	Uniform Elongation (UEL)/%	Total Elongation (TEL)/%
B650CL	615	723	9.4	21.5

## Data Availability

Not applicable.

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
