# Peer review of "A Novel Hydroforming Process by Combining Internal and External Pressures for High-Strength Steel Wheel Rims"

_materials, 2022, doi:10.3390/ma15196820_

Round 1

Reviewer 1 Report

Dear authors, Your article is very well prepared and suitable for publication after minor revisions. 

I recommend editing the following:

Line 111: the diameter of the initial tube is not apparent from Figure 2a. 

Line 224: What is the state of the initial material? Hardened, annealed, etc.? 

Check English grammar and spelling. 

Overall, the article deals with a very interesting topic. The technology of hydroforming is well known. However, the production of steel wheel rims using hydroforming could lead to a significant reduction in COX production while reducing the weight of car wheels by 1/3.

Reviewer 2 Report

I imagine and hope that this work will be very important for academia/materials science, based on the results presented, however, I believe that the greatest importance will be for the industry that may be interested in the idea presented by the authors since a reduction in the weight of the rim around of 13% is very significant and would lead to significant savings in the manufacturing process and in fuel consumption, which is important to note, there was not much emphasis in this work when we can relate to sustainability issues, but I understand that this was not the focus of the work.

For me, the article can be accepted for publication as long as it corrects or clarifies the following issues:

- In line 43, replace "apprearance" with " appearance";

- Apesar de Pe estar na Figura 3, um pouco antes ou depois da Equação 1, Pe não foi definido claramente como "pressão externa" no texto;

- Strangely line 147 is defined as "alfa", however, "alfa" don't appear in equations 1, 2, or 3. In my opinion, this definition should be after equation 5;

- Understanding that Pe is "external pressure" in Equation 1, why in Equation 5 are the authors using "Ps"???

- I suggest that be made the step by step to find the expression in Equation 5, because, for me, I think that have any wrong;

- In line 164 has "combining Equation 5 and 6 as", however, as combine to equation generate an inequation? Missing any information?

- In line 199, correct: "Equation 17" for "Equation 16".
